# Teacher Well-Being in Chinese Universities: Examining the Relationship between Challenge—Hindrance Stressors, Job Satisfaction, and Teaching Engagement

**DOI:** 10.3390/ijerph20021523

**Published:** 2023-01-14

**Authors:** Lan Xu, Jing Guo, Longzhao Zheng, Qiaoping Zhang

**Affiliations:** 1Research Center of Higher Education Development, Xiamen University, Xiamen 361005, China; 2Department of Mathematics and Information Technology, The Education University of Hong Kong, Hong Kong 999077, China; 3School of Journalism and Communication, Xiamen University, Xiamen 361005, China

**Keywords:** teacher well-being, challenge—hindrance stressors, teacher job satisfaction, teaching engagement, Chinese university teachers

## Abstract

Improving teacher well-being at work is a great challenge worldwide. Understanding the stressors of Chinese university teachers in teaching activities is critical for shedding light on well-being in the midst of the rapid expansion of the higher education system and the quest to rise in world rankings. This study integrates the well-being perspective and the transactional model of stress and coping to investigate the mechanisms underlying the effect of challenge—hindrance stressors on teacher engagement. Data were collected through the online platform SoJump in mainland China (N = 7743), and structural equation modeling was used to test the relationship between challenge—hindrance stressors and teaching engagement. The statistical results revealed the following: (1) challenge stressors had a significant positive effect on teaching engagement, while hindrance stressors were negatively related to teaching engagement; (2) challenge and hindrance stressors were significant negative predictors of teacher job satisfaction; (3) teacher job satisfaction suppressed the impact of challenge stressors on teaching engagement and partially mediated the process by which hindrance stressors impact teaching engagement. The findings suggest that the theoretically opposing effects of the two stressors are not absolute and that special consideration should be given to teachers’ job satisfaction in relation to stress management for university teachers.

## 1. Introduction

Teaching at a university is an extremely rewarding profession because the responsibilities of a university teacher are particularly rich, and the job of training people is challenging, but it also constitutes a variety of sources of stress. Kennedy pointed out that the academic responsibilities of university teachers include not only teaching, mentoring, discovering, publishing, telling the truth, serving the university, and reaching beyond the walls but also nurturing the next generation of scholars and adapting to future social changes [1]. Moreover, teaching in universities is different from that in primary and secondary schools. Not only is knowledge production academic, but its teaching is also complicated. Therefore, the concepts of “scholarship of teaching” and “scholarship of integration” are derived [2]. After the rise of “new public management” in a number of OECD countries in the 1980s [3], accountability has entered higher education and has made academics formally accountable for their performance [4]. Meanwhile, increasing levels of quality evaluation have led academics to feel that their professionalism is under attack [5]. University staff encounter challenges to their well-being, especially with growing work demands, lack of a work–life balance, and decreased work security [6,7]. High demands, fewer incentives, tremendous pressure, and fierce competition have all become labels associated with university teachers. Teacher well-being is a crucial issue for educational institutions and our society because of its close relation to teaching effectiveness, student outcomes, and educational governance [8,9]. As well as the distinctiveness of the academic profession, it is important to study the well-being of university teachers.

It is necessary to explain the background of the development of China’s higher education and the current situation of Chinese university teachers. First, in the past two decades, the absolute expansion of higher education in China has been unparalleled anywhere else around the world. The gross enrollment ratio of higher education reached 51.6% in 2019 [10], while in 1998, as the explosion began, this figure was 9.8% [11]. The goal of establishing research-targeting world-class universities focused on not only the expansion of the university but also the quality of teaching [12]. In 2004, the Higher Education Evaluation Center of the Ministry of Education was established to carry out a five-year evaluation of undergraduate teaching as a key measure to improve teaching quality. The institutionalization of this teaching evaluation made it a primary job for universities to cope with, and universities, of course, transferred this pressure to teachers. The former was required to increase investment in teaching funds, while the latter was required to increase the investment of teaching time and effort [13].

Second, with the gradual saturation of the academic labor market, the entry threshold and promotion criteria of Chinese universities are rising. This trend is spreading from the research universities with the goal of pursuing first-class teaching universities and local application-oriented undergraduate universities. Around the year 2000, Tsinghua University and Peking University began the reform of their personnel systems [14], implementing post-appointment and hierarchical mobility, and then introduced the tenure track system. The breaking of the “iron rice bowl”, “publish or perish” [15,16], and widening salary differences weakened the original stability and security of the academic profession and greatly increased the pressure and intensity of work [16,17].

Third, compared with Western universities, the administrative orientation of Chinese universities is obvious, which adds extra stress to the work of university teachers [18,19]. The teacher evaluation mechanism is dominated by administrative forces, which contradicts academic logic [20]. According to a series of survey results of Chinese university teachers, administrative matters were found to be the main variable for the increase in teachers’ stress compared to that in 2011 and 2018 [21]. There is a great conflict and contradiction between the view of the time required by academic work and the view of the time reflected by administrative management, which leads to crises concerning the physical and mental health, professional identity, and academic innovation of university teachers [22].

Job stress and job satisfaction have been the focus of international studies of the academic profession, which may impact teacher morale and work performance. For example, based on comparative survey (Changing Academic Profession, CAP) data, Teichler et al. drew the conclusion that, in most countries, the job satisfaction of academics is quite high on average, but academics from certain countries report below-average levels of satisfaction, such as in the UK, South Africa, and China [23]. Shin and Jung classified 19 countries into four types according to academic job satisfaction and job stress, showing that China has strong performance-based managerial systems, which are characterized by high stress and relatively low satisfaction. They also found that market-oriented managerial reforms are the main source of academic stress, while the high social reputation of academics in their society and academic autonomy are sources of job satisfaction [24].

Some studies confirmed that overwork and cumulative pressure could be harmful to university teachers’ health. In China, the overall physiological health of university teachers is deteriorating year by year, and stress from teaching, research, and administrative matters has been shown to have a significant positive impact on the detection of physiological disorders in university teachers to varying degrees [25]. Shen et al. found that occupational stress was a risk factor for depressive symptoms [26]. However, under the dominance of strong administrative forces, young teachers are trapped in the research output competition. They mainly deal with the time pressure by extending working hours, sacrificing leisure time and family life, and internalizing the consciousness of overwork [20]. In other words, the negative influence from the environmental pressure is, to some extent, covered up by the positive feelings brought by the drive of young teachers to pursue self-actualization.

Lazarus and Folkman’s transactional model of stress and coping provides a theoretical basis for challenge appraisal. They suggest that individuals tend to adopt positive action strategies when confronted with challenge stressors that may bring potential gains and negative strategies when they perceive the hindrance stressors to be harmful or potentially damaging [27]. Cavanaugh et al. developed a binary stressor scale based on this, suggesting that challenge stressors may have positive effects, and hindrance stressors have adverse effects [28]. Although some similar studies have been conducted in the Chinese context, they tend to focus on specific regions or disciplines [29,30]. There are few studies specifically discussing binary-work-stressor relationships with teaching engagement and job satisfaction.

Because of the features of the academic profession and the unique challenges faced by teachers in Chinese universities, it is necessary to carry out an investigation distinguishing challenge and hindrance stressors to understand how they affect teaching engagement and how positive emotions, such as job satisfaction, play a moderating role. Through this, we can identify effective ways to cope with work pressure, improve teachers’ work-related well-being, and achieve the purpose of improving teaching quality.

## 2. Literature Review and Hypotheses

### 2.1. Well-Being and Teachers’ Work-Related Well-Being

Well-being is a complex and multifaceted construct. In earlier studies, researchers referred to well-being as subjective well-being (SWB), including happiness, life satisfaction, and positive affect, which are influenced by health, social contact, activity, and personality [31]. SWB emphasizes individual emotion and proposes that well-being is an individual’s cognitive and affective evaluation of their life [32], p. 1. Ryan and Deci summarized two general perspectives: the hedonic approach and the eudaimonic approach to reconceptualize the term well-being [33]. Tov further elaborated on hedonic well-being (HWB) with affective well-being (AWB) and cognitive well-being (CWB) [34]. Some researchers distinguish between eudaimonic happiness and hedonic happiness, believing that eudaimonic well-being (EWB) makes people pursue meaning and value, making them more willing to participate in activities that allow for the actualization of one’s skills, talents, and potential [35]. Waterman reconsidered happiness from a eudaimonist perspective, proposed both conceptual and operational definitions of hedonic happiness and eudaimonic happiness, and analyzed four conceptions of well-being (subjective, hedonic, psychological, and eudaimonic) [36].

Work and organizational psychology compartmentalize the positive indicators (e.g., work engagement, happiness at work, and job satisfaction) and negative indicators (e. g., workaholism and burnout) associated with well-being in the work scenario [37]. In contrast, when studying an individual’s well-being in the work scenario, researchers not only focus on individual psychological satisfaction and positive emotional experiences but also tend to explore the actions that individuals have performed to enhance their well-being. Positive psychology emphasizes the role of well-being in personal growth and social development, and positive emotion, engagement, relationship meaning and purpose, and accomplishment are the five supporting elements of well-being [38], p. 34.

In recent years, university teachers’ well-being at work has attracted extensive research attention, but there has not been a specific definition of work-related well-being for university teachers. The common practice in previous studies is not to directly explain what well-being is but to use some related concepts to observe or measure it, such as work satisfaction [39,40] and work engagement [39,41]. Similar to work engagement, burnout, which is considered the opposite of engagement, was also used to measure the well-being at work of university teachers [42]. There are also studies on the influencing factors of university teachers’ work-related well-being. Teaching load, research demands, the new challenges demanded by modern educational technology and new technology and innovations [41], and the workload related to bureaucratic university practices [43] have been proven to have a negative impact on the work-related well-being of university teachers, while teaching resources, peer support, administrative support [41], and emotional job demands, such as maintaining relationships with colleagues and students [44], can increase their happiness to some extent. That is to say, unmet job demands and a lack of job support have become a source of stress [8], leading to threats to their well-being.

Well-being is particularly important for teachers since their work is characterized by strong self-actualization and altruism and is driven more by intrinsic motivation. They focus not only on their own professional development and the achievement of value goals but also on helping students to improve their personalities and promote their socialization. Because of the high social expectation of teachers’ personal virtues, it is particularly important to pay attention to teacher happiness. Only when they feel happy can they have better work performance and better help their students. Hence, in this study, we aimed to clarify the mechanism underlying the impact of work-related stress on university teacher teaching activities by investigating the relationship between stressors, teacher job satisfaction, and teaching engagement.

### 2.2. The Transactional Model of Stress and Coping

#### 2.2.1. Challenge—Hindrance Stressors

The transactional model of stress and coping can be used to frame the relationship between stressors, job satisfaction, and teaching engagement for university teachers. The model is shaped by three core concepts: stress is considered to emerge from the transactions between the person and the environment in the transactional model of stress and coping [45]. It occurs when an individual is unable to cope with the demands of the environment. Demand requires the individual to deal or cope with stress [8], leaving unmet demands to transform into the source of stress, called a stressor. Cognitive appraisal is the core of the transactional model of stress and coping, including primary appraisal and secondary appraisal [27]. Primary appraisal is considered to be the process of stressor categorization, focusing on what stressors mean to individuals. Although there are no “good” or “bad” stressors per se, individuals may label stressor attributes differently based on their primary appraisal of these stressors. Stressors that can offer potential gains have been classified as “challenges”, and those that can potentially be harmful or cause losses have been classified as “threats” [27], p. 33. Based on the characteristics of challenge and threat appraisal, stressors in the work context are classified as challenge and hindrance stressors. The prevailing view is that challenge stressors refer to job demands that are perceived to be beneficial for personal growth and development, such as workload, time pressure, job scope, and job responsibility, while hindrance stressors are those perceived as confinements to personal development and goal attainments, such as role ambiguity, lack of job security, red tape, organizational politics, and resource inadequacy [28].

Few researchers have confirmed whether stressors are a challenge or a hindrance for different participants, even though the challenge—hindrance stressor framework has been well-established in stress studies in various occupations. Horan et al. claimed that utilizing the challenge—hindrance stressor framework adopted by Cavanaugh et al. without reappraisal is inaccurate because differences in actual participants’ appraisal of stressors are ignored [46]. For example, time stress was found to be appraised as both challenging and hindering [47], which suggests that the use of challenge—hindrance stressors requires consideration of the different appraisals by the participants. Although researchers have used the challenge—hindrance stressor framework to study the stress profile of university teachers in mainland China, few researchers have considered the appraisal of both stressors by mainland Chinese university teachers [29,30]. Searle and Auton appraised stressors through items related to “judgments of expected impact on personal growth and/or achievement” [47], p. 126. Based on this criterion, several studies have been conducted piecemeal to appraise the stressors of university teachers in mainland China. For example, He et al. found that role ambiguity among university teachers in mainland China had a significant negative effect on job performance [48]. Han et al. found that a lack of work resources, including teaching resources, social support, and organizational support, would reduce the work engagement of teachers in university [29]. Although stressors were not specifically appraised in these studies, the above findings may provide some support for the applicability of the challenge—hindrance stressor framework to university teachers in mainland China.

The challenge—hindrance framework has now been demonstrated to provide theoretical support for secondary school teacher stress research [49], but there have been relatively few studies of university teachers. The line between challenge and hindrance stressors also does not seem to be as clear among university teachers. In some studies of university teachers, the effects of challenge and hindrance stressors have not always been reversed. For example, a study by Wu et al. found that both challenge and hindrance stressors were positively related to emotional exhaustion among Chinese university teachers [30]. Han et al. found that challenging job demands were positively associated with emotional exhaustion and negatively associated with work engagement, suggesting that the role of challenge stressors is not always negative [29]. This approach is not exactly the same as the widely used challenge—hindrance stressor framework. Therefore, there is a need to further explore whether there is also a distinction between challenge and hindrance stressors for university teachers and whether this framework can be applied to university teacher stress research.

#### 2.2.2. Teaching Engagement

Secondary appraisal, which is known as coping, refers to the cognitive and behavioral efforts that individuals make to meet the demands of work (in the face of stressors) [27]. Problem-focused and emotion-focused are the two main coping styles [50], with the former focusing on addressing the stressor itself and the latter dealing with the thoughts and feelings associated with the stressor. Pearsall et al. identified problem-solving as a positive coping strategy that corresponds to challenge stressors and avoidance coping as a negative coping strategy that corresponds to hindrance stressors [51]. In this study, the teaching engagement of university teachers will be used as an observable indicator of coping. The idea of teaching engagement was developed based on the concept of work engagement, which can be defined as the “harnessing of organization members’ selves to their work roles: in engagement, people employ and express themselves physically, cognitively, emotionally and mentally during role performances” [52], p. 694. Leiter and Maslach defined engagement from the perspective of positive psychology, suggesting that engagement is the opposite of burnout and proposing that engagement “consists of a state of high energy, strong involvement, and a sense of efficacy” [53], p. 94. Schaufeli et al. also acknowledged the opposite relationship between engagement and burnout, but they disagreed with using only degrees to distinguish between burnout and engagement and proposed that engagement is a separate concept to be measured in terms of vigor, dedication, and absorption [54]. Despite the definitions of work engagement being various, these definitions all contain an active tendency of “how employees treat their work.”

For teachers, Klassen et al. suggested that teaching engagement has unique characteristics. These authors provided a conceptual framework for teacher engagement, focusing more on classroom contexts and scenarios, which included four dimensions: cognitive engagement (CE), emotional engagement (EE), social engagement with students (SES), and social engagement with colleagues (SEC) [55]. In particular, the interaction between the teacher and other actors in teaching activities could not be omitted. A common understanding is that work engagement is reflected in individuals’ emotions, cognition, and behavior, and work engagement has a positive impact on work outcomes [56]. Leiter and Maslach also emphasized that the positive development of employees’ engagement is beneficial to their well-being and productivity [54]. Therefore, teaching engagement can be seen as positive coping with stressors. In terms of the definition, teaching engagement connotes a tendency for teachers to be proactive in their teaching; thus, in addition to considering the outcomes of teacher initiative, it would be a very interesting area of research to consider how to make teachers more proactive. As a result of the secondary appraisal, teaching engagement is likely to be influenced by stressors.

#### 2.2.3. Teacher Job Satisfaction

Lazarus and Folkman also suggested the role of emotion, proposing an explanatory framework for stress coping. They believed that challenge appraisal could trigger emotions, such as excitement and enthusiasm, while hindrance appraisal may bring about negative emotions, such as anger [27]. A widely used definition of job satisfaction is that job satisfaction is “a pleasurable positive emotional state resulting from the appraisal of one’s job or job experience” [57], p. 1304. The definition proposed by Locke supports the use of job satisfaction as an indicator of “emotion” in this study. The relationship between “emotion” and “coping” is also illustrated in the transactional model of stress and coping, where Lazarus and Folkman suggested that appraisal is accompanied by emotional responses; for example, challenge appraisals are often accompanied by enthusiasm, while hindrance appraisals may be accompanied by fear, and these emotions are thought to influence the individual’s choice of coping strategy [27]. This suggests that, in addition to being influenced by various stressors, teacher job satisfaction may also function as a dependent variable affecting teachers’ engagement in teaching.

Challenge—hindrance stressors, job satisfaction, and teaching engagement are all connected by the transactional model of stress and coping, which also provides a theoretical framework for investigating their interrelationship.

### 2.3. Challenge—Hindrance Stressors and Teaching Engagement

According to Lazarus and Folkman’s transactional model of stress and coping, stressors are stimuli that induce the stress process, and coping is the result of this process [27]. Coping is defined as the thoughts and behaviors that people use to manage internal and external stress-causing demands [27], p. 141. In the challenge—hindrance stressor framework, challenge stressors are believed to trigger positive coping, i.e., responding to challenges through problem-solving, and hindrance stressors may trigger negative and avoidant coping [27]. This suggests that stress will affect one’s performance and engagement in work. Challenge stressors have been shown to be significant and positive predictors of corporate employees’ work engagement, and hindrance stressors have produced the exact opposite effect [58]. Similarly, among university teachers, challenge stressors were found to be positively related to work engagement and exhaustion, while hindrance stressors were positively related to exhaustion and negatively to work engagement [59]. Accordingly, the following hypotheses were proposed:

**Hypothesis** **1a:**
*Challenge stressors positively predict teaching engagement.*


**Hypothesis** **1b:**
*Hindrance stressors negatively predict teaching engagement.*


### 2.4. Teacher Job Satisfaction as a Mediating Variable

The transactional model of stress and coping emphasizes the role of emotions in describing the relationship between stress and coping. It has been proposed that an individual’s cognitive assessment of stressors generates corresponding emotional responses and that these emotions influence the individual’s choice of coping strategy. Since job satisfaction is an employee’s subjective response to the work situation and is an emotional experience that arises when individuals evaluate their work experiences [57], many studies of the relationship between stress and work-related variables have used it as a mediating variable.

Teacher job satisfaction can be defined as the positive or negative evaluations that teachers make about their jobs [60]. It represents “a pleasurable emotional state resulting from the appraisal of a teacher’s job as achieving or facilitating their job values” [61], p. 139, and is often regarded as a positive indicator of well-being [29]. Many studies have demonstrated the negative impacts of university teachers’ work stress on their job satisfaction [62], but most of these studies did not determine whether the stressors are challenging or hindering. According to Greenglass and Burke [63], although teachers with high levels of stress may find fulfillment in their teaching careers, stress emanating from various sources may impact these teachers’ satisfaction levels. Past research has shown that challenge stressors tend to elicit positive affective responses that should more than offset the negative effects of strain [28]. This is because humans experience enjoyment and even euphoria when performing difficult and stressful tasks, even when they are aware that the demands may have a negative impact on their well-being [64]. When focusing on teachers’ work scenarios, there is no consensus argument on the results regarding challenge stressors. On the one hand, some empirical evidence has been reported showing that challenge stressors such as time pressures can decrease job satisfaction [65,66]. On the other hand, some studies have shown that faculty stressors related to factors intrinsic to teaching, including time pressures and wok-loads, have no significant effect on job satisfaction [67]. In this study, we include a lack of support, role ambiguity or conflict, organizational politics, and a lack of resources for professional development as hindrance stressors. In previous studies, role ambiguity has been shown to have a negative impact on job satisfaction of university teachers [67]. Wu et al. found that a greater amount of teaching resources, social support, and administrative support increased university teachers’ job satisfaction [30]. However, when teachers experience a lack of social support, communication with university authorities, training or financial assistance [67], a lack of status and promotion, and the management/structure of the school, this results in significantly associated job dissatisfaction [68]. In summary, teachers’ work is characterized by strong self-actualization and altruistic opportunities for personal growth and task accomplishment, and we assume that challenge stressors are positively associated with job satisfaction. Moreover, teachers perceive hindrance stressors to potentially endanger their personal growth and goal attainment; hence, we hypothesize that hindrance stressors are negatively connected with job satisfaction. Therefore, the following research hypotheses were formulated:

**Hypothesis** **2a:**
*Challenge stressors positively predict teacher job satisfaction.*


**Hypothesis** **2b:**
*Hindrance stressors negatively predict teacher job satisfaction.*


The relationship between job satisfaction and engagement among university teachers has been extensively studied, but most findings suggest that work engagement has a significant positive effect on job satisfaction [69]. Because individuals who are satisfied with their jobs are more motivated to achieve their job goals, which makes them more engaged in their work. The transactional model of stress and coping also emphasizes the positive impact of positive emotions on individual coping behaviors [27], so it is possible that, at a theoretical level, university faculty job satisfaction also has a positive impact on their job engagement. Furthermore, although few studies have directly examined the relationship between university teachers’ job satisfaction and teaching engagement, studies have demonstrated the positive effect of life satisfaction on individual performance in the workplace and in academic settings [70]. In summary, based on theory and the existing research findings, we propose the following hypotheses:

**Hypothesis** **3:**
*Teacher job satisfaction positively predicts teaching engagement.*


It has been established that job satisfaction among university teachers acts as a mediator in the relationship between a variety of variables. For instance, job satisfaction mediated the relationship between role stress, job performance, and health [42], and the relationship between job performance and a work–life balance [48]. However, few studies have looked at job satisfaction as a mediating factor between stressors and teaching engagement. Nevertheless, the transactional model of stress and coping suggests that job satisfaction, as an emotional response to stressors, influences individuals’ coping strategies [27,57]. The above-mentioned sorting of the relationship between the challenge—hindrance stressors, job satisfaction, and work engagement also appears to show that teacher job satisfaction mediates the effect of stressors on teaching engagement. On this basis, the following research hypotheses were developed:

**Hypothesis** **4a:**
*The relationship between challenge stressors and teaching engagement is mediated by teacher job satisfaction.*


**Hypothesis** **4b:**
*The relationship between hindrance stressors and teaching engagement is mediated by teacher job satisfaction.*


Figure 1 shows a conceptual model that was developed to address the relationship between challenge—hindrance stressors, teaching engagement, and teacher job satisfaction.

## 3. Methods

### 3.1. Participants and Procedure

This study was part of the National Survey on the Quality of Undergraduate Education in General Universities, commissioned by the Ministry of Education’s Supervisory Bureau and developed by the Ministry of Education’s Higher Education Teaching Assessment Center in mainland China. The expert team is composed of researchers from universities and research institutes, such as Xiamen University, Xi’an Jiaotong University, Tongji University, Beijing Academy of Educational Sciences, Guangxi Normal University, and Chizhou University, considering the type of institutions and the diversity of the geographical distribution. The national survey explores the quality of teaching in Chinese universities, including stakeholders’ perceptions of the teaching environment (including online teaching experiences), teaching self-efficacy, teaching outcomes, work stress, job satisfaction, and teaching engagement. Prior to the survey, the first draft of the questionnaire was designed in February, a workshop was held in March to determine the construct items, and data were collected through an online platform, SoJump (in Chinese, it is called Wenjuanxing), in April 2021. The questionnaire was sent to the national universities via the Ministry of Education’s Higher Education Teaching Assessment Center, which in turn pushed it to the academic affairs office, which forwarded it to the teachers. Based on convenience sampling, a total of 7978 participants responded to the survey. Participants who (1) answered the questionnaire in less than 8 min, (2) failed to pass the attention-check questions, (3) had an IP address that was duplicated or outside mainland China, and (4) responded with logical contradictions between the answers to the questionnaire were excluded from the data. Finally, 7743 responses remained, with an efficiency of 97.05%.

Respondents came from 210 universities in 21 provinces, 4 autonomous regions, and 3 municipalities across mainland China (from a total of 1245 higher institutes in 23 provinces, 5 autonomous regions, and 3 municipalities), covering 4 different levels of universities, including research universities (6.10%), teaching- and research-oriented universities (10.67%), and local application-oriented universities (83.23%), which is consistent with the overall regional and hierarchical distribution of higher institutions. There were 3177 male teachers and 4566 female teachers within the following geographical areas: East (30.48%), West (26.67%), Central China (29.52%), and Northeast China (13.33%). Most of them were between 31 and 60 years old (87.95%), with fewer aged 30 or below (10.25%) and 61 or above (1.80%). Most were university teachers from the natural science disciplines (49.26%), followed by the liberal arts disciplines (26.22%). Teachers from the social sciences were the smallest group (24.52%). The largest proportion of the sample was teaching- and research-oriented (56.17%), followed by teaching-oriented (41.53%), and the smallest proportion was research-oriented (2.30%). Therefore, the demographic characteristics of the sample are highly representative.

### 3.2. Instruments

The instruments used to measure each dimension were initially constructed in English. All members of the expert team have studied abroad and are fluent in English. Then, Chinese versions were developed for all the measures following the commonly used back-to-back translation procedure [71]. Further, the instrument was revised to reflect the consideration of the Chinese context as well as the research objectives, and the items were added and removed in response to comments from the Teaching Assessment Center.

#### 3.2.1. Challenge—Hindrance Stressors

Referring to Cavanaugh et al., a 12-item Likert scale was used to measure the participants’ challenge and hindrance stressors [28]. The scale was adapted to the Chinese university context through challenge stressors (e.g., the volume of assignments and essays that must be accomplished in the allotted time) and hindrance stressors (e.g., the degree to which my teaching career seems “stalled”). Higher average scores indicated higher levels of challenge and hindrance stressors (1 = not at all; 5 = fully; Cronbach’s α _Cha_ = 0.88, Cronbach’s α _Hin_ = 0.91; M_Cha_ = 3.12; SD_Cha_ = 1.03; M_Hin_ = 2.61; SD_Hin_ = 1.05).

#### 3.2.2. Teacher Job Satisfaction

Adapted from the scale of Ho and Au [72], a 3-item Likert scale was used to measure the extent of teachers’ job satisfaction (e.g., working as a teacher is extremely rewarding), with higher average scores indicating higher levels of teacher job satisfaction (1 = not at all; 5 = fully; Cronbach’s α = 0.91; M = 4.22; SD = 0.89).

#### 3.2.3. Teaching Engagement

Adapted from Klassen et al., a 17-item Likert scale was adapted to the Chinese context to measure teaching engagement [55]. The scale retained the same four dimensions of cognitive engagement (e.g., I try my hardest to perform well while teaching), emotional engagement (e.g., I feel happy while teaching), social engagement with students (e.g., in class, I care about the problems of my students), and social engagement with colleagues (e.g., At school, I value the relationships I build with my colleagues), with higher average scores indicating higher levels of teaching engagement (1 = not at all; 5 = fully; Cronbach’s α = 0.95; M = 4.60; SD = 0.51).

### 3.3. The Strategy of Data Analysis

SPSS 25 and structural equation modeling with Amos 21 were used to investigate the relationships among the challenge—hindrance stressors, teacher job satisfaction, and teaching engagement. With the aid of Amos 21, we followed a two-step approach to test the measurement and structural models [73]. The first step was to evaluate the measurement model, and the second step was to use 5000 samples to generate bias-corrected confidence intervals for each coefficient in this procedure.

## 4. Results

### 4.1. Assessment of the Measures

We performed a confirmatory factor analysis (CFA) as the first step to certify the measurement model. First, CFA was performed to analyze the goodness of fit of the measurement model. The results showed an acceptable fit to the data (χ^2^/*df* = 35.03, *p* < 0.001, NNFI = 0.942, CFI = 0.953, TLI = 0.927, RMSEA = 0.066, RMR = 0.046). The NFI, CFI, and TLI all reached the acceptable standard of 0.9 [74]. Moreover, the RMSEA was 0.066, which is less than 0.08, suggesting this model is well-fit.

Table 1 shows that all the composite reliability (CR) values for all the latent variables were between 0.88 and 0.91, which is higher than the threshold value of 0.70 [75]. Factor loadings and the average variance extracted (AVE) were used to assess the convergent validity. Almost all measurement items were higher than 0.60, and the AVE values exceeded the cut-off value of 0.50 [75]. Therefore, all the constructs indicated satisfactory convergent validity. In order to verify discriminant validity, we compared the square roots of the AVE with the correlations among the latent variables [76]. The square roots of the AVE for each construct were greater than the correlation coefficients. This confirmed the discriminant validity of the latent variables. The square roots of the AVE values and the correlation coefficients of the constructs are reported in Table 2.

In order to address the issue of common method bias, we conducted Harman’s single-factor test. The results showed that four factors emerged from the factor analysis with eigenvalues greater than 1. The variance explained by the largest factor was 36.93%, which was less than 40% [77].

### 4.2. Hypothesis Testing

In the second step, the model was further used for hypothesis testing (Figure 2). We used 5000 samples to produce bias-corrected confidence intervals for each coefficient in this procedure. We evaluated the predictability of the structural model through the variance *R*^2^ in the values of all the dependent latent constructs. This explained about 16% of the variability in teacher job satisfaction and 29% of the variability in teaching engagement. Then, we assessed ƒ², representing the effect size of the exogenous latent variables. The calculations show that the effect size of the challenge stressors on teacher job satisfaction and teaching engagement was 0.012 and 0.005, respectively; the effect size of hindrance stressors on teacher job satisfaction and teaching engagement was 0.047 and 0.006, respectively, and the effect size of teacher job satisfaction on teaching engagement was 0.304.

Hypothesis 1 predicted that (a) challenge stressors are positively related to teaching engagement and (b) hindrance stressors are negatively related to teaching engagement. Our results showed that the challenge stressors (*β* = 0.057, *p* < 0.01) and hindrance stressors (*β* = −0.045, *p* < 0.01) significantly influenced teaching engagement positively and negatively, respectively.

Hypothesis 2 posited that (a) challenge stressors are positively related to teacher job satisfaction, and (b) hindrance stressors are negatively related to teacher job satisfaction. Our results showed that the challenge stressors (*β* = −0.113, *p* < 0.01), and hindrance stressors (*β* = −0.201, *p* < 0.01) negatively influenced teacher job satisfaction. Therefore, Hypothesis 2 was partially supported.

Hypothesis 3 states that teacher job satisfaction is positively related to teaching engagement. The results showed that teacher job satisfaction significantly influenced teaching engagement (*β* = 0.348, *p* < 0.01). Thus, Hypothesis 3 was supported.

Hypothesis 4 tested whether teacher job satisfaction mediates the effect of (a) challenge stressors and (b) hindrance stressors on teaching engagement. Teacher job satisfaction had a significant partial mediating effect among challenge stressors (*β* = −0.065, *p* < 0.01, 95% CI = [−0.087, −0.043]) and hindrance stressors (*β* = −0.158, *p* < 0.001, 95% CI = [−0.182, −0.136]) on teaching engagement. Previous studies proposed that, in the mediation model, when the direct and indirect effects are opposite, this indicates a suppression effect in the model [77]. The direct effect of challenge stressors on teaching engagement was 0.057, and the indirect effect was −0.066; the mediating role (i.e., teacher job satisfaction) greatly suppressed the effect of challenge stressors on teaching engagement. Teacher job satisfaction had a significant partial mediating effect between hindrance stressors and teaching engagement, accounting for 76.70% of the total effect. Thus, Hypothesis 4a and Hypothesis 4b were both supported.

## 5. Discussion

This study provided empirical evidence for the application of the transactional model of stress and coping in a higher education context. From the well-being perspective, it tested the two ways that Chinese university teachers deal with stress in response to challenge—hindrance stressors, and examined the influence of stress on their well-being from two aspects: job satisfaction and teaching engagement. The results support our hypothesis that challenge stressors and hindrance stressors, respectively, have a significant positive or negative impact on teaching engagement. Meanwhile, this study found that job satisfaction suppressed the effect of challenge stressors on teaching engagement and partially mediated the effect of hindrance stressors on teaching engagement.

### 5.1. Theoretical Implications

On the one hand, this study confirmed that hindrance stressors are negatively related to teacher job satisfaction and teaching engagement, and teacher job satisfaction partially mediates the role of hindrance stressors and teaching engagement. Hindrance stressors that are seen as obstacles to personal growth and goal achievement [28] can directly negatively affect teachers’ emotional state (i.e., job satisfaction) and work performance (i.e., teaching engagement), which is consistent with the previous literature on hindrance stressors and emotional exhaustion and performance [30,48]. Moreover, hindrance stressors are negatively associated with teachers’ job satisfaction, which in turn negatively predicts their teaching engagement. Previous research has shown that university teachers with negative characteristics such as low self-efficacy and competition avoidance will tend to adopt avoidance-coping strategies to escape the possible threat of stress [78]. It was proposed that when hindrance stressors cause pressure, hindrance appraisals may be accompanied by fear [27], which can frustrate the emotional state at work and reduce teaching engagement due to low job satisfaction. This finding is also supported, to some extent, by the theory of conservation of resources, which states that individuals tend to maintain, protect, and construct to satisfy their demands [79]. Stress arises in work situations when individuals need to consume resources to meet job demands. When teachers perceive that their workplace is not beneficial to their teaching activities or personal development but rather a drain on their existing resources, they may choose avoidance coping strategies to conserve resources, i.e., reduce their teaching engagement.

On the other hand, this study demonstrated that challenge stressors had a positive impact on teaching engagement, which echoes a previous study about challenge stressors that were believed to positively predict work engagement [58]. This is because challenge stressors are energizing and provide opportunities for feelings of accomplishment, as well as growth and development [28,29]. Challenge stressors are job requirements that are thought to be helpful for professional and personal development [28], which suggests that stressors can enhance teachers’ teaching engagement to positively cope with stressors, as Leiter and Maslach noted that positive development of engagement is beneficial to employee well-being and productivity [54]. Those who are goal-oriented and willing to persevere in their pursuit of personal growth will benefit from the challenge stressors [80]. In order to encourage university teachers to engage in their work, it is appropriate to present them with some challenging work demands during the management process of higher education. However, we obtained the opposite result in terms of the relationship between challenge stressors and teachers’ job satisfaction [81], i.e., challenge stressors had a significant negative effect on teacher job satisfaction, as did the hindrance stressors. Surprisingly, when teacher job satisfaction was considered in the impact model of the challenge stressors on teaching engagement, it was found that teacher job satisfaction significantly suppressed the effect of challenge stressors on teaching engagement.

More specifically, in this model, the total effect of the challenge stressors on teaching engagement was not significant when teachers’ job satisfaction was considered. The possible explanation for this is that, despite the fact that the stressors associated with challenges can be energizing and provide opportunities for fulfillment as well as growth and development [28,29], they can also have a dark side. Challenge stressors can even negatively affect job satisfaction when stress is excessive and unresolved. As teachers’ work is characterized by strong self-actualization and altruism [35]; it takes a long time and effort to engage in teaching, and it is difficult to feel rewarded in the short term, which leads to reduced job satisfaction. When confronted with challenge stressors that bring delayed but lasting rewards, teachers will evaluate their own ability (whether their teaching ability is outstanding) and whether external support is sufficient to help them overcome difficulties (such as financial support, professional development training, and technical support services). When the conditions are met, teachers will be extremely satisfied and eager to engage in teaching. If the conditions are not met, the challenge stressors will become a risk that may result in loss or frustration. To summarize, while challenge stressors have a positive impact on teaching engagement, once the teacher’s tolerance is exceeded, job satisfaction suppresses the effect of challenge stressors on teaching engagement. This suggests that the positive effect of challenge stressors on teaching engagement is not absolute and that challenge stressors may play a different role when other factors (i.e., job satisfaction) enter the model.

### 5.2. Practical Implications

The results of this study not only support the application of the transactional model of stress and coping in the context of higher education in China, but also reveal some ways to improve the well-being of university teachers in practice, including increasing job satisfaction and promoting their teaching engagement.

First, this study distinguishes challenge—hindrance stressors, especially recognizing the complex impact of challenge stressors on the well-being of university teachers, indicating that university teachers need to adopt effective coping strategies to cope with increasing stresses and highlighting the importance of a comprehensive understanding of academic duties. The workload and time pressure of university teachers should be controlled in a stimulating and achievable range. On the one hand, universities should encourage teachers to treat challenge stressors sources as opportunities for growth rather than obstacles, emphasizing that overcoming challenges will bring teachers a sense of achievement, but also should not ignore the potential adverse effects of challenge stressors on teachers’ happiness. Therefore, a teacher’s work burden and time pressure should be regulated according to the actual situation and controlled within a stimulating and achievable range, which should avoid placing infinite pressure on teachers, reduce the obstructive pressure, and release the potential of teachers. At the same time, managers should also reassess the emotional stimulation of teachers caused by different stressors, provide teachers with healthier patterns of emotional regulation strategies [44], and directly challenge stress in a positive direction instead of toward negative burnout. On the other hand, teachers should identify the goals of their professional development in teaching, research, and service and clarify the links between stress and their fundamental responsibilities [41]. Academic responsibilities should be clarified to avoid the ambiguity of roles, and trials unrelated to teaching should be reduced so as to help teachers better allocate their time and improve work efficiency.

Second, the significant relationships between job stress, job satisfaction, and job engagement revealed in this study imply the need to improve teachers’ job satisfaction. When combined with the hypothesis of the transactional model of stress and coping, this study shows that satisfaction plays a mediating role in the direction of challenge—hindrance stressors. Although the high pressure of teaching combined with other responsibilities causes some teachers to overwork themselves, challenge stressors, such as innovative pedagogies, can still promote teachers’ teaching input if the teaching work can offer them a high reputation, organizational recognition, and other satisfaction [82]. Previous research has rarely used satisfaction as a mediator variable, but satisfaction as an emotional experience of happiness should not be overlooked. In China, an important reason for the low satisfaction of teachers is that the evaluation system focuses on research rather than teaching [21,83]. The orientation of research as a priority will cause the behavior choice of university teachers to deviate from teaching [83,84]. However, to increase academic productivity with limited energy, teachers can only put in more effort, and regard the continuous delay of working hours as “self-exploitation” [20], even sacrificing time for physical exercise. In this case, it is particularly necessary for administrators to change the supremacy of administrative logic and respect the academic logic of teachers’ work [85]. Giving more preferential support to teaching in policy-making means changing the undiversified and quantitative evaluation system and improving the recognition of teaching engagement.

Third, it is recognized that teaching engagement should be regarded as an important factor in improving university teachers’ overall well-being, and creating a supportive and cooperative environment for teaching is effective in improving teachers’ well-being. According to the results of this study, teaching engagement is reflected in the positive attitudes and emotions in teaching, the guarantee of teaching time for preparation and course design, teachers’ sense of efficacy, and the improvement of interpersonal relationships with students and colleagues. To cope with the challenges of new ideas, new methods, and new technologies, teachers need to master more teaching skills. Without adequate support, teachers will feel frustrated, engage in dissatisfaction and escape behaviors, and not try to innovate in teaching. This highlights the importance of university administrators providing appropriate support. The university should increase teaching resource input, initiate and reward teaching innovation, and enhance the availability and sharing of teaching resources [13]. Administrators need to plan effectively for faculty development, not only to address improvements in teaching tasks such as curriculum design, teaching strategies, and learning assessment but also to encourage university teachers to communicate and cooperate with others. Teachers should have a comprehensive understanding of professional development and master stress management strategies such as practicing mindfulness and release techniques, rearranging work schedules, and maintaining a balance between work and family life to enable them to adopt effective coping strategies in an increasingly stressful environment.

### 5.3. Limitations

Both the challenge—hindrance stressor framework and the transactional model of stress and coping of teachers have been validated in Western scenarios, and recently, Chinese university teachers have attracted the attention of researchers. The results of this study have confirmed some of the hypotheses but not all of them. This suggests that it is valuable to promote Western stress theory in China but with an emphasis on Chinese teachers’ characteristics. However, this study has some limitations. First, the measurement of the challenge—hindrance stressors and teaching engagement was based on the teachers’ self-perceptions of stress and engagement. Although the reliability and validity of both the stressor scale and the teaching engagement scale were tested with good results, there may be personal differences in the perceptions of university teachers. For instance, the degree of their perception could also affect the data of challenge—hindrance stressors and teaching engagement, which in turn could affect the performance of these factors in the model. Follow-up studies may attempt to further test the correlation between the variables and further verify the relationship by excluding subjective individual differences. Second, the results of this study suggested that challenge and hindrance stressors only have opposing effects on teaching engagement. The applicability of the challenge—hindrance stressor framework to university teachers requires further validation. Third, the questionnaire was mainly distributed by the Higher Education Evaluation Center of the Ministry of Education, whose duty is to conduct undergraduate teaching assessments, which may have resulted in the influence of challenge—hindrance stressors on the teaching engagement of university teachers being confounded by external factors (e.g., top-down administrative pressure), resulting in low self-reports of challenge—hindrance stressors and relatively concentrated teaching engagement in the data collected, further leading to the small beta values.

## Figures and Tables

**Figure 1 ijerph-20-01523-f001:**
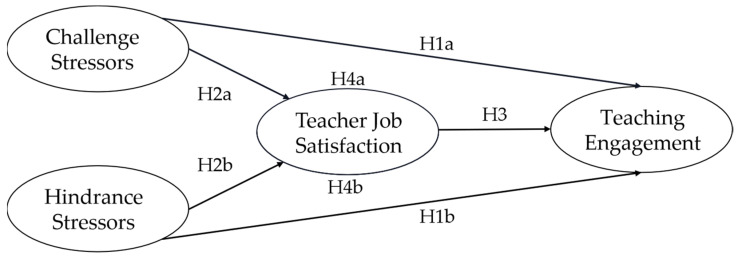
Conceptual model of the relationship among challenge—hindrance stressors, teacher job satisfaction, and teaching engagement.

**Figure 2 ijerph-20-01523-f002:**
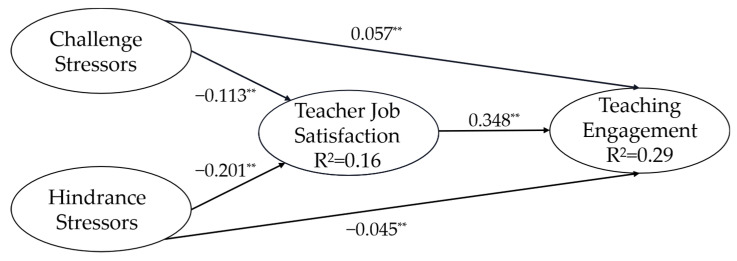
Structural equation model of challenge—hindrance stressors, teacher job satisfaction, and teaching engagement. Note: ** *p* < 0.01, N = 7743.

**Table 1 ijerph-20-01523-t001:** Results of the measurement model assessing construct reliability (N = 7743).

	Items	Factor Loading	CR	AVE
CS1	The volume of homework and essays that must be corrected in the allotted time.	0.793	0.88	0.55
CS2	The non-teaching related responsibility of my position entails.	0.778		
CS3	The number of research projects and/or assignments I have after work time.	0.703		
CS4	The amount of time I spend at teach.	0.770		
CS5	The tight teaching schedule and time pressures I experience.	0.772		
CS6	The new teaching technologies, ideas, skills, and innovations I have to keep up with.	0.618		
HS1	The degree to which my teaching career seems “stalled”.	0.857	0.91	0.63
HS2	The underestimation of my Job-related abilities.	0.776		
HS3	The lack of social support at work I have.	0.856		
HS4	The feeling of isolation I have.	0.659		
HS5	The lack of communication with administrators about the expectation of my teaching.	0.803		
HS6	The lack of funding or training to develop my teaching potential.	0.782		
TJS1	I enjoy working as a teacher.	0.791	0.91	0.77
TJS2	If I had to choose a career again, I would choose a teacher.	0.913		
TJS3	Working as a teacher is extremely rewarding.	0.929		
TE1	I carefully design the course outline/syllabus.	0.777	0.95	0.55
TE2	I carefully design the teaching program.	0.794		
TE3	I prepare appropriate teaching materials for each lesson.	0.807		
TE4	I try my best to perform well while teaching.	0.773		
TE5	I spent time investigating and analyzing students’ learning conditions.	0.695		
TE6	In class, I try to understand students’ standpoints.	0.812		
TE7	In class, I pay attention to students’ feelings.	0.848		
TE8	In class, I care about the problems raised by students.	0.822		
TE9	I focus on motivating students to learn.	0.830		
TE10	I listen to students’ opinions and suggestions in teaching.	0.824		
TE11	I encourage students to participate in classroom activities in a variety of ways.	0.817		
TE12	I feel happy while teaching.	0.583		
TE13	I am excited about teaching.	0.589		
TE14	I love teaching.	0.611		
TE15	At school, I value my relationship with my colleagues.	0.614		
TE16	At school, I am committed to helping my colleagues.	0.665		
TE17	At school, I care about the problems of my colleagues.	0.626		

Note: CS = challenge stressors; HS = hindrance stressors; TJS = teacher job satisfaction; TE = teaching engagement.

**Table 2 ijerph-20-01523-t002:** Mean, standard deviation, and correlation matrix (N = 7743).

	Mean	SD	1	2	3	4
1. Challenge stressors	3.115	1.028	0.74			
2. Hindrance stressors	2.607	1.049	0.685 **	0.79		
3. Teacher job satisfaction	4.216	0.889	−0.325 **	−0.367 **	0.88	
4. Teaching engagement	4.571	0.508	−0.149 **	−0.225 **	0.515 **	0.74

Note: ** *p* < 0.01 (two-tailed). CS = challenge stressors; HS = hindrance stressors; TJS = teacher job satisfaction; TE = teaching engagement. The values bolded on the diagonal are the AVE square root of each construct.

## Data Availability

Restrictions apply to the availability of these data. Data were obtained from the project team commissioned by the Chinese Ministry of Education and are available from the corresponding authors with the permission of the Chinese Ministry of Education. This process can be initiated upon request to the corresponding author.

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
