# Peer review of "Teacher Well-Being in Chinese Universities: Examining the Relationship between Challenge—Hindrance Stressors, Job Satisfaction, and Teaching Engagement"

_ijerph, 2023, doi:10.3390/ijerph20021523_

Round 1

Reviewer 1 Report

This is an interesting study and well-constructed.  There is no issue with the structure and the coherence. The analysis is carried out satisfactorily, using the correct statistical applications.  

However, I would like to bring up some issues, though minor:

1. Proofread the whole manuscript carefully to weed out spelling mistakes.  Such as found in the abstract - (analysed); page 2 - l.86, 98 to name a few.

2. How do you come up with the five categories of No stress; mild stress; moderate stress, high stress and severe stress?  Ned to explain that.

Author Response

Many thanks for taking the time to review this manuscript and providing valuable comments. We really appreciate all your comments and suggestions! We have carefully considered the comments and suggestions and made some changes. We have uploaded a copy of the original manuscript with all the changes highlighted by using the track changes mode in Word. Appended to this letter is our point-by-point response to the comments raised by the reviewers. The comments are reproduced, and our responses are given directly in blue colour.

Reviewer 2 Report

Dear authors,

The issue of instructor psychological well-being in Chinese higher education is of great importance and interest, particularly given reforms to instructor evaluation in recent years. This study has potential in providing some implications, however, I do believe that there are several critical points that should be first considered. The major points will be listed first, followed by some comments by section:

Major comments:

1. The research model and hypothesized paths require further empirical support and logical argumentation to justify the present investigation. In particular, the mediating effects of job satisfaction should be more strongly supported in the context of stressors and work/teaching engagement. Without support, from related studies, there is insufficient evidence for the proposed mediation.

2. Several constructs require more adequate elaboration and definition in the literature review, with better support from relevant studies. For example, challenge-hindrance stressors in the context of education (particularly higher education) require more elaboration and development in the literature review. The adoption of the transactional theory of stress would require a more detailed and specific approach to the framing of research questions and hypotheses. Other theories mentioned, such as the Job Demand-Resources Theory or Conservation of Resources theory may have some explanatory power, if adopted systematically. I suggest reading some of the thoughts shared in the following article: Horan, K. A., Nakahara, W. H., DiStaso, M. J., & Jex, S. M. (2020). A review of the challenge-hindrance stress model: recent advances, expanded paradigms, and recommendations for future research. Frontiers in Psychology, 11, 560346.

3. The literature cited must be interpreted correctly and elaborated upon sufficiently, in context. Several citations do not fit with the claims they are associated with. For example, please check the following citations and determine whether or not they are appropriate in context:

[3] - More from this systematic review could be included, such as the need for evaluating other teacher well-being related outcomes, such as effectiveness, teaching quality, and impact on student outcomes.

[4] – not a study on educators

[5] - Not a suitable source. This "introduction to a special issue" is referring to the studies of Catano et al. (2010) and Winefield et al. (2003). These original sources are not cited in your paper.

[6] - Too general. Citing an entire book for a broad point such as this is not appropriate.

[7] – Evaluates a sample of high school students

[8] – Includes both staff and students in HE

[9] – This study is related to cognitively enhancing prescription drug use

[12] - Measured anxiety and depression as hypothesized outcomes of work stress. The context needs to be clearer here.

[10] – On page 2: This study does not mention either physical health or depression.

[14] - Does not assess the link from stress to either physical health or depressive symptoms.

[15] - Does not include physical health or depressive symptoms.

[16] - The study also discusses well-being as measured by cognitive-affective compounds, job-related affect, and context-free affect.

[19] - Not available in English.

[20] - Not clearly linked to stress.

[21] - Not focused on teaching duties alone. More support is needed.

[22] - The abstract states "most of the participants viewed research and teaching as mutually supportive"

[23] – The source is unavailable.

[24] - The role of coping in the transactional model deserves more attention here.

[26] – The sample was secondary school teachers

[37]: - Not about teachers

[38]: - Elementary and secondary level teachers

[39] - Primary, secondary, and vocational schools

[40] - Does not clearly refer to workload-related stress.

[41] - Be more precise. The TALIS is a large-scale survey.

[42] - More details are needed here. The study evaluated burnout as an aspect of teacher well-being, not as a related variable.

[43] - The study is not relevant to job satisfaction.

[44] - Well-being is not the focus of this study.

[45] - These two variables are part of the construct of "job resources" in the job demands-resources model. In the study they are positively associated with engagement. However, the variables (and job resources) are not associated with well-being in the model

[46] - In the study, stress is only associated with burnout.

[47] - An opinion article, not an empirical study.

[48] – Does not clearly fit in the context (page 3)

[50, 51] – These studies are lumped together, but the findings should be presented more clearly and precisely

[60] - Why would the focus only be on teaching, not on research and service-extending, as these are mentioned earlier as vital factors in teacher stress and well-being.

[67] - Focuses on hindrance stress factors.

[68] - This study describes complicated relationships related to time pressure and job satisfaction, including the role of "challenge appraisal"

[69] – In this study, management/structure of the school’ and ‘lack of status and promotion’ predicted job dissatisfaction.

[70] – This study discusses supervisor-reported performance. Is this a match to teaching engagement?

[71] - Not clearly related to engagement

[72] - No mediation was evaluated in this study.

4.  Some portions of the literature review provide claims that are not supported with citations. Please check the following line numbers: 25, 46, 48-49, 156, 189, 206-207, 209-210.

5. In other places, there needs to be better clarification of certain terms or constructs. These include “extensibility” (line 38), “health conditions” (lines 40-41), “different countries” (line 43), “up-or-not tenure track” (line 56), “work performance” (line 194).

6. There is not sufficient information provided on the instruments (1. How was the scale validated more specifically? 2. What is the reliability (internal consistency)? 3. Sample items? How can the scale be evaluated in terms of specificity to the teacher population?)

7. Results and discussion of the effects of gender, age, and professional title are provided, but there was no mention of these factors in the literature review. Therefore, their inclusion in the analysis is not warranted based on the current content of the manuscript.

8. There are no effect sizes reported for the findings. As such, although there may be significant differences, it is very difficult to interpret whether or not these are meaningful.

9. The research procedures need to be described in more detail, including the thresholds for CFA criteria and the procedures for SEM, including the evaluation of mediation effects.

10. The fact that the mean values for teacher’s job satisfaction and engagement are very high suggests that the analytical results may be impacted. As such, analysis of individuals with relatively lower levels of job satisfaction or engagement may be required in order to evaluate the findings adequately in order to provide a clear and meaningful interpretation.

11. Several portions of the Discussion section are speculative in nature, not based on the data, not in line with the literature review, or not supported by suitable citations. Examples of parts of the Discussion section that are problematic include the following line numbers: 397-400, 402-421, 429-435. 441-446.

12. The conclusions should be worded with greater care since, as noted, teachers tended to report high levels of job satisfaction and engagement. Additionally the potential of job satisfaction as a moderator should be considered, particularly since there is very little support for mediation in the literature review, or the relationship between job satisfaction and engagement.

13. There are no clear implications. Some general suggestions in the conclusions are provided, but there are few specific and actionable suggestions. Engagement with more current literature related to strategies for addressing higher education instructors’ stress is recommended.

Comments by section

Overall:

- Few parts need more paraphrasing, as too similar to original source. (Lines 25-36, 96-97, 100-106, 172-174)

Abstract:

- More information on the nature of the instruments for data collection could be provided.

- The nature of direct and indirect effects could be clarified.

Keywords:

- Are university-level instructors referred to as “teachers” or “instructors” or “faculty members/staff” This is worth consideration here and throughout the manuscript.

Introduction:

- How working in higher education is an “extremely rewarding profession” is not clearly stated

- Line 40: The article [10] evaluates mental health. You need to be specific here.

- Line 43: Specify the countries

- Line 46: No citation. The "dying of illness or sudden causes" is not elaborated upon here or linked to stress by any empirical evidence.

- More on the Chinese context could be provided to develop a more clear research gap/rationale.

- Lines 67-69: It is not clear how your study is in-depth. No qualitative data was collected and the data were analysed without considering an evaluation of respondents by profile. In terms of “work stress,” “engagement, “and satisfaction” -These three elements have been researched quite extensively. How the Chinese context is different would need to be more clearly defined. Also some of this research would need to be used to support the development of a research model.

- Line 74: for “positive effects,” be specific that this is in reference "work performance" However this finding was in the context of other variables in a complex model

- More support is needed for Cavanaugh et al.’s scale, since it is claimed that it was based on Lazarus and Folkman. In fact, the contribution of the following studies appears to have been more influential: Judge, T. A., Boudreau, J. W., & Bretz, R. D. (1995). Job and life attitudes of male executives. Journal of Applied Psychology, 79, 767-782.  Bretz, R. D., Boudreau, J. W., & Judge, T. A. (1994). Job search behaviour of employed managers. Personnel Psychology, 47, 275-301.

- Line 86: How does your study specifically evaluate the “mechanisms underlying” the impact of stressors on teaching.

Literature review:

- Line 98: It is stated that there are two approaches for conceptualizing well-being, but this is an oversimplification. In fact, Waterman [citation 36] mentions "four conceptions of well-being (subjective, hedonic, psychological, and eudaimonic)"

- Lines 129-134: Needs more nuance in the writing, including some descriptions of the variables being mentioned.

- Lines 152-153: This gap needs to be better developed.

- Wording of hypotheses should not include “may”

- Line 194: Can “work performance” be convincingly compared to “engagement.” These are separate constructs.

- Line 196: More work needs to be done to integrate the construct of “burnout” in the context of your research

- Lines 217 and 218: H2a and H2b are not sufficiently supported in terms of “challenge” and “hindrance” stressors based on the preceding discussion and citations.

Method:

- The “expert team” qualifications should be stated more specifically

- The qualifications of the translation team should be specified

- The influence of COVID-19 in terms of the data collected and the subsequent analysis and interpretation should be evaluated.

- The sampling frame, data collection strategy is not stated in sufficient detail

- How was the representativeness of the sample evaluated (for example, based on population statistics)?

- More precise exclusion criteria could be provided.

Results:

- The wording on lines 303-304 is incorrect. You cannot say that challenge stressors caused higher stress. You can only report that teachers reported higher levels of challenge-related work stress compared to hindrance-related work stress.

- As mentioned above, effect sizes are required for all statistical tests

- In Table 2 and elsewhere, attention to the number of significant digits in reporting results is needed.

- On lines 346-350 there is mention of some elements related to the stressors as evaluated by your instruments. However, since the information on the instruments, including the items included, was not provided in sufficient detail, the reader is not able to draw clear conclusions on interpretations.

- Lines 380-384: Explanation of partial mediation is needed. What was the direct vs. mediated effects overall?

Discussion:

- The wording of lines 398-400 is very difficult to understand. Moreover, the claim that challenge stressors caused more stress is not accurate based on the data collected. It can only be reported that respondents reported higher levels of challenge stressors relatively speaking.

- Line 405: The discussion of work-family conflicts is speculative. Does this fit the Chinese context, as the papers are not from domestic sources?

- Lines 406-409: This discussion is not warranted as there were no hypothesized gender effects.

- Lines 410-421: 1. Interpretation cannot be supported by the nature of the data available.

2. Not included as a research hypothesis, so the discussion is not warranted. If this is to be included, there needs to be some literature in the introduction/lit review to develop a hypothesis.

- Lines 426-427:  The conclusion stated here is not reasonable when worded like this since the word "reduce" suggests intention. This is not clearly supported by your previous literature review.

- Lines 429-431: Not clearly relevant.

- Line 434: There is no clear evidence of “avoidance coping strategies” based on the data collected.

- Lines 441-444: A comparison of the direct and indirect effects and more appropriate discussion or interpretation is required. As satisfaction and engagement are different constructs, it is possible that multiple underlying mechanisms are at play. Also, given the high means for both job satisfaction and engagement, it is unclear how a "negative" effect would be interpreted. The data would need to be scrutinized further, such as through evaluation of high vs. low job satisfaction or engagement individuals. The potential of moderating effects vs. mediating effects should be considered. Job satisfaction may serve as a moderator for the relationship instead. This was not evaluated.

- Lines 446-451: This is not phrased in a manner suggestive of "teacher engagement"

- Line 452: What does this mean?

- Lines 456-466: not based on the data. A limitation must be introduced here. Not enough data was collected to evaluate these kind of relationships. E.g., the job demands-resources model

Conclusion:

- Lines 470-472: Difficult to support, given high job satisfaction and engagement.

- Lines 474-475: How is this supported by the data?

- Lines 482-484: too general. Use the literature to provide better implications.

References:

- Some of the articles cannot be accessed.

- In several cases the DOI addresses provided are incorrect.

Overall, I hope that the above suggestions are helpful for you in revising your manuscript, if the journal makes that decision. Best wishes in your current and future research.

Author Response

(The authors gave the same response as above.)

Round 2

Reviewer 2 Report

The revisions made by the authors have improved the quality of the manuscript. I have a few remaining concerns that I hope can be addressed:

1. Effect sizes should be reported and the small beta values should be mentioned in the limitations section.

2. The research gap should be more strongly stated before the literature review.

3. Actionable implications should be provided with appropriate citations from empirical studies (Discussion and Conclusion). There is still not enough in terms of clear contributions to theory and practice.

4. The content on page 14 is lacking in empirical support through appropriate citations.

5. My previous critique of the use of citations should be re-examined, since several have not been appropriately revised, for example the following:

[3] (now [8]) - More from this systematic review could be included, such as the need for evaluating other teacher well-being related outcomes, such as effectiveness, teaching quality, and impact on student outcomes.

[6] (now [11]) - Too general. Citing an entire book for a broad point such as this is not appropriate.

[17] (now [18]) - Fisher is not advocating this as a statement that can be generalized, but as a "common sense" belief held by "lay people."

[20] (now [21]) - Not clearly linked to stress.

[21] (now [23]) - Not focused on teaching duties alone. More support is needed.

[22] (now [24]) - The abstract states "most of the participants viewed research and teaching as mutually supportive"

[37] (now [38]) - Not about teachers

[38] (now [38]) - Elementary and secondary level teachers

[39] (now [40]) - Primary, secondary, and vocational schools

[40] (now [41]) - Does not clearly refer to workload-related stress.

[41] (now [42]) - Be more precise. The TALIS is a large-scale survey.

[42] (now [43]) - More details are needed here. The study evaluated burnout as an aspect of teacher well-being, not as a related variable.

[43] (now [44]) - The study is not relevant to job satisfaction.

[44] (now [45]) - Well-being is not the focus of this study.

[45] (now [46]) - These two variables are part of the construct of "job resources" in the job demands-resources model. In the study they are positively associated with engagement. However, the variables (and job resources) are not associated with well-being in the model

[46] (now [47]) - In the study, stress is only associated with burnout.

[47] (now [48]) - An opinion article, not an empirical study.

[48] (now [49]) – Does not clearly fit in the context (page 3)

[67] (now [73]) - Focuses on hindrance stress factors.

[68] (now [74]) - This study describes complicated relationships related to time pressure and job satisfaction, including the role of "challenge appraisal"

[69] (now [75]) – In this study, management/structure of the school’ and ‘lack of status and promotion’ predicted job dissatisfaction.

Best wishes in your research!

Author Response

We are very grateful to you for reviewing the manuscript carefully. The comments are very helpful for improving the quality of our manuscript. We have carefully considered the comments and suggestions and made some changes accordingly. Please see the attachment.
